# Dimethylformamide Inhibits Fungal Growth and Aflatoxin B_1_ Biosynthesis in *Aspergillus flavus* by Down-Regulating Glucose Metabolism and Amino Acid Biosynthesis

**DOI:** 10.3390/toxins12110683

**Published:** 2020-10-29

**Authors:** Lin Pan, Peng Chang, Jing Jin, Qingli Yang, Fuguo Xing

**Affiliations:** 1Institute of Food Science and Technology, Chinese Academy of Agricultural Sciences/Key Laboratory of Agro-products Quality and Safety Control in Storage and Transport Process, Ministry of Agriculture, Yuanmingyuan West Road, Haidian District, Beijing 100193, China; panlin@caas.cn (L.P.); jinjing@caas.cn (J.J.); 2College of Food Science and Engineering, Qingdao Agricultural University, Qingdao 266109, China; czhhaonugai@126.com (P.C.); yql@qau.edu.cn (Q.Y.)

**Keywords:** aflatoxins, *Aspergillus flavus*, dimethylformamide, mechanism, glucose metabolism, amino acid biosynthesis

## Abstract

Aflatoxins (AFs) are secondary metabolites produced by plant fungal pathogens infecting crops with strong carcinogenic and mutagenic properties. Dimethylformamide (DMF) is an excellent solvent widely used in biology, medicine and other fields. However, the effect and mechanism of DMF as a common organic solvent against fungal growth and AFs production are not clear. Here, we discovered that DMF had obvious inhibitory effect against *A. flavus*, as well as displayed complete strong capacity to combat AFs production. Hereafter, the inhibition mechanism of DMF act on AFs production was revealed by the transcriptional expression analysis of genes referred to AFs biosynthesis. With 1% DMF treatment, two positive regulatory genes of AFs biosynthetic pathway *aflS* and *aflR* were down-regulated, leading to the suppression of the structural genes in AFs cluster like *aflW*, *aflP*. These changes may be due to the suppression of VeA and the subsequent up-regulation of FluG. Exposure to DMF caused the damage of cell wall and the dysfunction of mitochondria. In particular, it is worth noting that most amino acid biosynthesis and glucose metabolism pathway were down-regulated by 1% DMF using Kyoto Encyclopedia of Genes and Genomes (KEGG) analysis. Taken together, these RNA-Seq data strongly suggest that DMF inhibits fungal growth and aflatoxin B_1_ (AFB_1_) production by *A. flavus* via the synergistic interference of glucose metabolism, amino acid biosynthesis and oxidative phosphorylation.

## 1. Introduction

*Aspergillus flavus* as plant-invasive fungal pathogens cause enormous losses in the yield and quality of field crops worldwide [1]. Under the suitable environmental conditions, *A. flavus* is prone to produce a series of strong carcinogenic and mutagenic secondary metabolites aflatoxins (AFs) during the process of infecting food and feed [2]. AFs are the predominant and most carcinogenic naturally occurring compounds which inevitably result in health complications, including hepatocellular carcinoma, acute intoxication, immune system disorder and growth retardation in children [3,4]. In 1993, AFs were classified as a Class I carcinogens by the International Agency for Research on Cancer (IARC) [5,6]. Among AFs, aflatoxin B_1_ (AFB_1_) is the most toxic and carcinogenic compound known. AFs are commonly relevant to the cereals, nuts and a scope of their agricultural products, especially peanuts, maize and rice [7,8]. In the United States, the Food and Drug Administration (FDA) has set the limiting value at 20 μg/kg for total AFs (B_1_, B_2_, G_1_, G_2_) for all foods, and 100 μg/kg for peanut and corn feed products [9,10]. In the European Union, the European Commission (EC) set the upper limit at 2 μg/kg for AFB_1_ and 4 μg/kg for total AFs [11]. The appearance of antifungal resistance of chemical fungicides and the safety requirements of practical application in crops globally have incurred the discovery of novel antifungal agents and other antifungal substances [12].

Many strategies have been used to reduce AFs contamination. Safe and efficient natural substances for preventing and controlling *A. flavus* growth and AFs production are necessary. Essential and plant hormone possessing potent anti-microbial, antioxidant activities, were applied to agricultural industry [13,14]. However, some fungicides have low solubility in general solvents. As a universal solvent, dimethylformamide (DMF) is effective to dissolve several high-efficiency antifungal agents. Therefore, to determine the inhibitory effect and mechanism of DMF on *A. flavus* is necessary for basic research. Although DMF have toxicity, previous research showed only inhibited the cell viability in cells exposed to 160 mM DMF (78.7%, *p* < 0.01) [15].

Transcriptional sequencing (RNA-Seq) has been widely applied to study lots of eukaryotic transcriptomes [16,17,18]. It also has been used to decipher the inhibitory mechanism of eugenol [19], gallic acid [20], and cinnamaldehyde [21] on *A. flavus* growth and AFs formation. The objective of this study is to investigate the effect of DMF on *A. flavus* growth and AFs production, and to determine transcriptomic changes in *A. flavus* treated with DMF. In particular, the inhibitory mechanism of action of DMF on AFs biosynthesis at the transcriptomic level is elucidated. In terms of agricultural applications, this research may provide a basis for synergistic antifungal effect between DMF and new fungicides.

## 2. Results

### 2.1. Inhibitory Potential of DMF Acts on Growth and Toxicity by A. flavus

The antifungal effect of DMF on *A. flavus* is shown in Figure 1. After DMF treatment, the mycelia growth of *A. flavus* was significantly suppressed in a dose-dependent manner. Under 4% DMF concentration, the maximum growth inhibition was observed (Figure 1A). However, the complete inhibition was not obtained with all the tested concentrations. With 0.25–1% of DMF treatment, the colony growth was retarded compared to the control group (Figure 1B).

Similarly, as shown in Figure 2, DMF significantly inhibited the AFB_1_ production in a dose-dependent manner in the yeast extract sucrose (YES) broth at the tested levels. Moreover, the production of AFB_1_ was completely suppressed by 2% and 4% DMF. Obviously, the difference compared with the growth data, DMF has a significant inhibitory effect on AFB_1_. Taken together, these results indicated that DMF significantly suppressed *A. flavus* growth and AFB_1_ production in a dose-dependent manner.

### 2.2. Changes on Gene Expression Profile of A. flavus Treated with DMF

The transcriptomes of all treatment and control groups were sequenced to obtain a comprehensive overview of the transcriptional response of *A. flavus* to DMF. Using RNA sequencing, averagely 46.35 million, 48.23 million raw reads were gained from control and 1% of DMF treatment samples, respectively. After percolating, about 44.39 million and 46.39 million clean reads were obtained from control and treatment transcriptome. According to fragments per kilobase per million mapped fragments (FPKM) values with FDR ≤ 0.05 and log_2_Ratio ≥1 or ≤−1, differentially expression genes (DEGs) between the control and treatment samples were identified. Compared with control, a total of 2353 genes were significantly differentially expressed in the DMF group. Among them, 1204 (51.17%) genes were up-regulated and 1149 (48.83%) genes were down-regulated. These results suggested that DMF effectively influenced the expression of large number of genes.

### 2.3. Functions and Involved Pathways of Significant DEGs

GO functional enrichment analysis revealed that these significantly differentially expressed genes were mainly involved in molecular function (MF), cellular component (CC), and biological process (BP). Figure 3 shows the top 30 terms of the most obvious enrichment in three gene categories. For the up-regulated genes (Figure 3A), flavin adenine dinucleotide binding, inorganic molecular entity transmembrane, metal ion transmembrane transporter activity were the staple terms in molecular function. Lipid catabolic process and mental iron transport were the most affected terms belonging to the biological process, followed by signaling, signal transduction and intracellular signal transduction. The main terms in cellular component were Golgi-associated vesicle membrane, coated vesicle membrane, vesicle membrane, cytoplasmic vesicle membrane, vesicle coat. The result indicated that the treatment with 1% DMF mainly up-regulated the genes involved in the vesicle and related cellular component. However, most of down-regulated genes were enriched in biological process (BP) (Figure 3B). For the down-regulated genes, organic acid metabolic process, oxoacid metabolic process, carboxylic acid metabolic process, small molecule biosynthetic process and cellular amino acid metabolic process were the abundant terms in biological process. Oxidoreductase activity, protein dimerization activity, NAD binding and electron transfer activity were the predominant terms in molecular function. The most terms belonging to cellular component were mitochondrion, mitochondrial part, organelle envelope and envelope.

By Kyoto Encyclopedia of Genes and Genomes (KEGG) pathway analysis, the top 20 enriched pathways of significant DEGs in *A. flavus* treated with 1% DMF treatment were shown in Figure 4. For up-regulated DEGs with 1% DMF (Figure 4A), the most abundant genes (26 DEGs) were enriched protein processing in endoplasmic reticulum (afv04141), and 20 DEGs, 16 DEGs, 13 DEGs, 11 DEGs were enriched in RNA transport (afv03013), MAPK signaling pathway (afv04011), phenylalanine metabolism (afv00360), beta-alanine metabolism (afv00410), respectively. For down-regulated DEGs (Figure 4B), the most down-regulated genes (72 DEGs) were enriched in biosynthesis of amino acids (afv01230), and 38 DEGs, 31 DEGs, 14 DEGs were enriched in carbon metabolism, 2-Oxocarboxylic acid metabolism (afv01210) and valine, leucine and isoleucine biosynthesis (afv00290). In general, the most important amino acid (such as valine, leucine, isoleucine, arginine and lysine) biosynthesis pathways were suppressed, while some amino acid (alanine, phenylalanine) metabolism and degradation pathways were improved. In addition, the genes involved in fungal basal metabolism (carbon metabolism and nitrogen metabolism) and mitochondrial were also down-regulated.

### 2.4. Genes Related to Pigment Biosynthesis and Fungal Development

Transcriptional activity of genes involved in *A. flavus* pigment and development was presented at Table 1. The genes related to pigment biosynthesis AFLA_016120 (O-methyltransferase family protein), AFLA_016130 (a hypothetical protein), AFLA_016140 (conidial pigment biosynthesis scytalone dehydratase *Arp1*) were all significantly up-regulated. Obviously, consistent with the growth phenotype with exposure to 1% DMF treatment, most gene involved in fungal development were not significantly influenced. A FluG family protein (AFLA_039530), a conidiation-specific family protein (AFLA_044790) and a conidiation protein Con−6 (AFLA_044800) was conspicuously stimulated. APSES (ASM-1, Phd1, StuA, EFG1, and Sok2) transcription factor *stuA*, developmental regulator *flbA*, C_2_H_2_ conidiation transcription facto *flbC* and *brlA* were up-regulated slightly. However, sexual development transcription factor *steA*, sexual development transcription factor *nsdD*, G-protein complex alpha subunit *gpaA/fadA* were down-regulated. LaeA which regulated secondary metabolism was slightly down-regulated. Interestingly, the developmental regulator VeA show slight down-regulated level, as the velvet regulator VosA which plays a pivotal role in spore survival and metabolism in *Aspergillus* [22] was tightly up-regulated.

### 2.5. Genes Related to the Biosynthesis of Aflatrem, Aflatoxins, and Cyclopiazonic Acid

The transcription activities of the genes involved in the biosynthesis of aflatrem (#15), aflatoxins (#54), and cyclopiazonic acid (#55) were shown in Table 2. In pathway #15, most of genes were expressed at very low levels, and the expression of AFLA_045470 and AFLA_045530 was undetected. It is worth mentioning that AFLA_045570 encoding a putative MFS multidrug transporter and AFLA_045460 encoding a putative acetyl xylan esterase were significantly down-regulated. However, most of other genes were slightly up-regulated, including the genes encoding the hybrid PKS/NRPS enzyme, integral membrane protein and cytochrome P450. In AFs biosynthesis pathway (54#), all genes were down-regulated by with 1% DMF except *aflNa* encoding a hypothetical protein and *aflA* encoding fatty acid synthase alpha subunit were slightly up-regulated. The key regulator genes *aflR* and *aflS* were both slightly suppressed with log_2_ D1/CK values −0.263 and −0.419, respectively. Interestingly, all genes referred to lipid redox were significantly down-regulated, including *aflYa*, *aflY*, *aflX*, *aflW*, *aflP*, *aflO*, *aflM*, *aflJ* and *aflH.* In Pathway 55, AFLA_139470 encoding a FAD dependent oxidoreductase, AFLA_139480 encoding a tryptophan dimethylallyl transferase and AFLA_139480 encoding a hybrid PKS/NRPS enzyme were significantly up-regulated with 1% DMF treatment, whereas AFLA_139460 encoding an MFS multidrug transporter was obviously down-regulated.

### 2.6. Genes Involved in Cell Wall

The transcription activities of the genes involved in cell wall were shown in Table 3. AFLA_038420 encoding an endo-chitosanase B and AFLA_024770 encoding a putative symbiotic chitinase were significantly up-regulated with log_2_ D1/CK values 4.6754 and 2.2492, respectively. Rather than, the genes related to chitin hydrolysis and chitin synthesis were both significantly down-regulated. It is worthy that the down-regulated transcriptional levels of gene involved in chitinase were much higher than chitin synthase. All genes related to glucan synthesis were up-regulated, AFLA_023460 encoding an alpha−1,3-glucan synthase Ags1 and AFLA_134100 encoding an alpha−1,3-glucan synthase Ags2 were up-regulated with log_2_ D1/CK values 1.8014 and 0.7643, respectively. All genes related to glucan hydrolysis were significantly down-regulated, including AFLA_095680 encoding a putative alpha−1,3-glucanase, AFLA_029950 encoding a putative endo−1, 3(4)-beta-glucanase, AFLA_045290 encoding a putative extracellular endoglucanase/cellulase, AFLA_102640 encoding a putative exo-beta−1, 3-glucanase, AFLA_053390 encoding a GPI-anchored cell wall beta−1,3-endoglucanase EglC, AFLA_068300 encoding a 1, 3-beta-glucanosyltransferase Bgt1.

### 2.7. Genes Involved in Glucose Metabolism Pathway

The transcription activities of the genes involved in glucose metabolism pathway was shown in Appendix A. All genes involved in glucose metabolism were down-regulated at different degrees. In glycolysis pathway, AFLA_101470 encoding a putative glyceraldehyde−3-phosphate dehydrogenase, AFLA_085400 encoding a 2,3-bisphosphoglycerate-independent phosphoglycerate mutase, AFLA_069370 encoding a putative phosphoglycerate kinase PgkA and AFLA_119290 encoding a putative phosphofructokinase were significantly down-regulated with log_2_ D1/CK values −5.1528, −1.1066, −1.0872 and −1.0641, respectively. In the tricarboxylic acid (TCA) cycle and glyoxylic acid cycle, AFLA_052400 encoding a isocitrate lyase AcuD, AFLA_086400 encoding a putative isocitrate dehydrogenase Idp1 and AFLA_069370 encoding a putative phosphoglycerate kinase PgkA were obviously down-regulated with log_2_ D1/CK values −1.6033, −1.1747 and −1.0872, respectively.

### 2.8. Genes Involved in Oxidative Phosphorylation and Amino Acid Biosynthesis/Metabolism

AFs biosynthesis may be controlled by the energy state of specific subcellular compartments, and the production of these secondary metabolites may be affected by the synthesis of ATP in mitochondria [23]. As shown in Figure 5, there are five complexes involved in the oxidative phosphorylation, including complex I, II, III, IV, and V. In each complex, several genes encoding NADH dehydrogenase, succinate dehydrogenase, cytochrome oxidase and ATPase, were down-regulated at different degrees by 1% DMF. In addition, the biosynthesis pathways and metabolism pathways of almost all 20 essential amino acid were affected by 1% DMF, such as phenylalanine, tryptophan, tyrosine, valine, leucine, isoleucine, alanine, glycine, serine, threonine, and cysteine. It is noteworthy that most of amino acids metabolism related genes were up-regulated, while most of amino acids biosynthesis genes were down-regulated in *A. flavus* treated with 1% DMF. It indicates that the amino acids cannot be synthesized, then some important intermediate products cannot be accumulated with putting a lot of pressure on the fungal cells.

### 2.9. Genes Involved in MAPK Pathway, Oxylipins, GPCRs and Oxidative Stress Response

As shown in Appendix A, the expression changes of most genes in oxidative stress response (OSR), MAPK pathway, oxylipins and GPCRs were slightly changed after 1% DMF treatment. The spore-specific catalase CatA was significantly up-regulated, while catalase Cat was suppressed. For MAP kinase, the *sakA1* and *sakA2* were both up-regulated by DMF with Log_2_ (FPKM) values 0.678 and 1.837, respectively. The fatty acid oxygenase *ppoA*, *ppoB* and *ppoC* were all significantly up-regulated. Interestingly, *gfdB* encoding a glycerol 3-phosphate dehydrogenase, was clearly down-regulated. The G protein-coupled receptor gene *gprG* encoding a PQ loop repeat protein was significantly down-regulated after DMF treatment, while other receptor genes *gprH*, *gprK*, *gprM* were significantly up-regulated.

## 3. Discussion

The biosynthesis of toxic and carcinogenic AFs involves multiple biochemical reactions, which require the activity of more than various 27 enzymes [24]. These enzymes are coded by the genes grouped in a cluster of aflatoxin pathway, as their expression regulated by cluster-specific transcription activator *aflR* and transcription enhancer *aflS* [25,26]. After DMF treatment, all genes in the cluster were down-regulated at different degrees except for *aflA* encoding fatty acid synthase alpha subunit. However, DMF could not completely inhibit any genes in the cluster. Two crucial regulator genes *aflS* and *aflR* were slightly repressed in *A. flavus* with DMF treatment, accompanying with significant reduction in the expression of AFs structural genes such as *aflD*, *aflH*, *aflI*, *aflJ*, *aflL*, *aflM*, *aflO*, *aflP, aflQ*, *aflW*, *aflX* and *aflY*, leading to a consecutive loss in the ability to synthesize AFs intermediates [24]. These findings suggest that the expression changes of structural genes are more significant compared with the key regulators *aflR* and *aflS* with inhibitor treatment.

Similar findings were obtained in *A.flavus* treated with eugenol [17,19], piperine [27], cinnamaldehyde [21,28], ethanol [29], 5-Azacytidine (5-AC) [30] and gallic acid [20]. In *A. flavus* treated with 5-AC, the expressions of *aflR* and *aflS* were basically unchanged. However, at least three structural genes including *aflQ*, *aflI* and *aflLa* were completely inhibited, and five structural genes were suppressed with high or middle levels by 5-AC, especially *aflG* and *aflX* [30]. After treatment with eugenol, the expression of *aflR* did not change obviously and the expression of *aflS* was slightly up-regulated. The expression of most structural genes was down-regulated and the most strongly down-regulated gene was *aflMa*, followed by *aflI*, *aflJ*, *aflCa*, *aflH*, *aflNa*, *aflE*. However, the expression of these genes was not completely inhibited [31]. Exposed to cinnamaldehyde, *aflR* and *aflS* showed slightly up-regulation. Excepting the up-regulated expression of *aflF*, all the structural genes in the cluster were down-regulated. The most strongly down-regulation gene was *aflD*, followed by the key structural genes *aflG*, *aflH*, *aflP*, *aflM*, *aflI*, *aflL* and *aflE* [21]. When *A. flavus* treated with antioxidant gallic acid, *aflR* and *aflS* were slightly up-regulated while structure gene showed down-regulation [20]. After treatment with 3.5% ethanol, *aflR*, *aflS* were all down-regulated significantly and *aflS*/*aflR* showed the up-regulation. At least three structural genes including *aflK*, *aflLa*, *aflL*, *aflG* and *aflM* were completely inhibited following the up-regulation of *aflS*/*aflR* [28]. These similar results showed that after treatment with different anti-aflatoxigenic compounds, the AFs cluster transcription factors *aflR* and *aflS* were not significantly changed although many structural genes were significantly repressed, suggesting the stable expression of the two key regulator genes in *A. flavus*.

Sugars metabolism produce the basic substance unit acetyl-CoA, is prerequisite for all known polyketides compounds especially AFs [32,33]. Acetyl-CoA is mainly generated by glycolysis pathway in cytoplasm and fatty acid β-oxidation pathway in peroxisomes [34,35]. The reduction of AFs biosynthesis was associated with the decrease of tricarboxylic acid (TCA) cycle intermediates, the suppression of fatty acid biosynthesis, and the increase of pentose phosphate pathway substance, as reported [36]. The numerous DEGs were enriched in carboxylic acid metabolic process. The utilization of a given carbon source in *A. flavus* involves the sugar cluster, of which *aflYe*, *aflYd*, *aflYc*, *aflYb* were up-regulated. Additionally, a zinc finger transcription repressor *creA* was slightly down-regulated, *creA* suppresses the expression of *aclR* and the latter is a positive regulatory factor for the genes *aldA* [37,38]. It is showed that the DMF treatment contributes to the significant increase of pentose phosphate pathway of genes expression, such as AFLA_079220 (glucose dehydrogenase), AFLA_026950 (3-ketoacyl-CoA thiolase peroxisomal A precursor) from our research. The similar result was reported that gallic acid inhibited the AFs formation via up-regulation of pentose phosphate pathway [20]. Cinnamaldehyde, cultured in AFs inhibitory medium, the pentose phosphate pathway was accelerated, leading to NADPH accumulation and AFs reduction [21,28]. On the other hand, a large number of genes involved in TCA cycle-related and glycolysis pathway showed significant down-regulation. Interestingly, the glyoxylic acid-related genes especially AFLA_052400 encoding an isocitrate lyase AcuD, AFLA_049390 (malate synthase AcuE) and the TCA cycle-related genes especially AFLA_086400 (socitrate dehydrogenase Idp1) were significantly inhibited, leading to the accumulation of isocitrate and the subsequent depletion of acetyl-CoA. Taken together, the inhibition of AFs may be due to the depletion of acetyl-CoA and the lack of NADPH.

One important factor that has been found to affect AFs biosynthesis is amino acid catabolism and biosynthesis [39]. Wilkinson et al. [39] reported that glutamine and tyrosine favor AFs production in *A. flavus*, while tryptophan seem more complicated. Adye et al. and Naik, M et al. [40,41] reported that tyrosine, tryptophan, phenylalanine and methionine were easily absorbed into AFs biosynthesis pathway of *A. flavus*. Similarly, Wilkinson et al. [39] found the supplementation of amino acid in YES media could positively modulate the AFs biosynthesis in *A. flavus* and *A. parasitiucs*. In addition to these aromatic amino acids which easily influenced the AFs biosynthesis, Payne et al. [42] unearthed that proline and asparagine can increase more AFs production than tryptophan or methionine in *A. flavus*. Roze et al. [43] found that *veA* negatively regulated catabolism of branched chain amino acids and the synthesis of ethanol in *A. parasiticus*. The analogous results existing in our study were that *veA* was slightly suppressed, and the up-regulated and down-regulated DEGs were abundantly enriched in metabolism and biosynthesis pathways, respectively. However, 2-PE at a low level rendered the decreased activities in the metabolism of branched-chain amino acid, of which may be necessary to activate the AFs pathway by providing building blocks and energy regeneration [44]. These results implied that amino acids played complex roles in AFs production.

Oxidative phosphorylation in mitochondria can convert the energy released by organisms into ATP during the decomposition process. Chung et al. [45] reported that DMF influenced electron-transfer reactivity of cytochrome b5. As a powerful inhibitor of electron transport, DMF was found to have a marked inhibiting effect of the phosphorylation reaction at concentrations lower than 6.0% (*v*/*v*) [46]. Consistent with the result, our RNA-seq data showed that the expressional levels of several genes in oxidative phosphorylation were consistently down-regulated, including complex I (NADH complex), complex II (dehydrogenase complex), complex III (cytochrome complex) and complex IV (cytochrome oxidase). All genes related to electron transport component were repressed at different degrees, especially AFLA_129610 encoding a putative subunit G of NADH-ubiquinone oxidoreductase was obviously down-regulated. Inhibition of oxidative phosphorylation by the application of exogenous compounds such as resveratrol has been shown to compromise fungal oxidative stress tolerance by altering mitochondrial respiration and oxidative stress as a prerequisite for AFs production by *Aspergillus parasiticus* [47,48]. These results suggested that oxidative phosphorylation dysfunction might be associated with the reduction of AFs biosynthesis.

The cell wall acts as a protective barrier for fungi against environmental factors and is essential for the survival of the fungus during development and reproduction [49]. The cell wall plays an important molecular target for various antifungal compounds [50]. The important components of fungal cell wall, alpha−1,3-glucan and beta−1,3-glucan, play a crucial role to maintain the normal morphology of fungal cell wall. The alpha−1,3-glucan synthase encoded by *ags*1, *ags*2, *ags*3 is essential for the formation of alpha−1,3-glucan [51]. *fksP* encoding the beta−1,3-glucan synthase participates in the synthesis of beta−1,3-glucan [52]. In our RNA-seq data, *ags*1, *ags*2, *ags*3 and *fksP* were significantly up-regulated to resist DMF destruction. As the important constituent ingredient of the cell wall, chitin biosynthesis is directly controlled by chitin synthase [53]. In our current study, all chitin synthase was significantly depressed with DMF treatment. Chitinase facilitates the separation of its cells during fungal growth and reproduction [54]. Wang et al. [55] found the glucanase gene *crh*11 was down-regulated, could cause the obstruction of fungal reproduction. Similarly, our RNA-seq data displayed that all glucanase genes related to cell wall were down-regulated at different degrees. These results suggest that DMF attacks the cell wall of *A. flavus*, with destroying the main components of the fungal cell wall, chitin and structural polysaccharides. Consequently, the cell will generate a stress response to maintain the basic structure of the cell wall by over-expressing alpha/beta-1,3-glucan synthase genes to resist external stimuli.

Figure 6 showed an elementary diagram illustrating the antifungal effect of dimethylformamide act on *A. flavus* NRRL3357. To sum up in Figure 6, dimethylformamide inhibits the AFs biosynthesis and fungal growth of *A. flavus* via (1) attacking the cell wall by regulating the expression of cell wall integrity (CWI) related genes, and then cause the disorder of related protein of the cell membrane, and the spread to oxylipins genes by signal transduction; (2) increasing the depletion of acetyl-CoA and suppressing the NADPH accumulation by glucose metabolism; (3) disturbing the function of oxidative phosphorylation, then reducing ATP deemed as key elements for fungal cell to perform various reactions; (4) weakening amino acid synthesis which are indispensable for AFs accumulation. In brief, the dysfunction of cell wall integrity, glucose metabolism, amino acid biosynthesis, oxidative phosphorylation tight-knit interact with AFs production.

## 4. Conclusions

This study provides new insights of mechanism to interpret the inhibition of transcriptional regulation with DMF against AFs synthesis via a large number of comparative RNA sequencing. Based on existing research, we conclude that (1) DMF attacks the cell wall of *A. flavus*, with destroying the fungal cell wall integrity and the cell will subsequently generate a stress response to maintain the basic structure of the cell wall by over-expressing glucan synthase genes to resist external stimuli; (2) in the presence of DMF, the most intuitive performance of the decrease of AFs production is following increased expression of their specific regulators *aflS/aflR* and down-regulation of AFs cluster genes; (3) the down-regulation of the global regulator VeA and up-regulation of FluG is associated with the increase of conidiophore development; (4) glucose metabolism pathways are greatly interfered including TCA cycle, pentose phosphate pathway, glyoxylic acid pathway; (5) oxidative phosphorylation is in disorder, with reducing ATP required by fungal organisms to perform various reactions; and (6) the biosynthesis and metabolism of most amino acid is affected, this indicates that most amino acids cannot be synthesized and some important intermediate products cannot be accumulated, which puts a lot of pressure on the fungal cells. In general, these results strongly suggest that DMF disturb a variety of cellular reactions in *A. flavus*, thereby interfering fungal growth and metabolic function.

## 5. Materials and Methods

### 5.1. Fungal Strain, Chemicals and Treatment

DMF (DMF, 100% purity) was purchased from Beijing Chemical Works (Beijing, China). Chromatographic grade methanol was purchased from Thermo Fisher Scientific (Waltham, MA, USA). The AFB_1_ standard was purchased from Sigma-Aldrich (Sigma-Aldrich Chemicals, St. Louis, MO, USA).

The *A. flavus* strain NRRL3357 used in this study [29] was maintained in the dark condition on potato dextrose agar (PDA) medium at 4 °C as reserving. A conidia inoculum was prepared by washing PDA surface culture and adjusted to 10^7^ conidia/mL with 0.1% Tween-80 solution. The AFB_1_ standard was dissolved in 70% methanol.

### 5.2. Determination of Fungal Growth and AFB_1_ Production

Different treatment of DMF was added to the sterilized PDA medium at final concentrations of 0.25%, 0.5%, 1%, 2%, and 4% respectively. Then, 10 µL of 10^7^ conidia/mL suspension was inoculated on the central of PDA medium and incubated at 28 °C for 7 days. Determination of *A. flavus* growth indexes were by measuring colony diameters.

Similarly, the different treatment concentrations of DMF were added to yeast extract sucrose (YES, Hopebio, Qingdao, China) broth to obtain the concentrations of 0.25%, 0.5%, 1.0%, 2.0% and 4.0% DMF. The control cultures (CK) were treated similarly but without DMF. Then, 100 µL of 10^7^ conidia/mL suspension was inoculated to 100 mL YES broth. Fungal mycelia were collected and weighed as the method described by Yamazaki et al. [56] after incubation at 28 °C and 180 rpm in the dark for 5 days. Extraction and quantification of AFB_1_ by high-performance liquid chromatography (HPLC) was conducted according to the previous reference [57]. AFB_1_ was extracted with acetonitrile: water (84:16) solution from YES broth and purified by a ToxinFast immunoaffinity column (Huaan Magnech Biotech, Beijing, China). An Agilent 1220 Infinity II system coupled with a fluorescence detector (Santa Clara, CA, USA), an Agilent TC-C_18_ column (250 mm × 4.6 mm, 5 μm particle size) and a post-column derivation system (Huaan Magnech, Beijing, China) was used to quantify the AFB_1_ concentrations. Each treatment was performed in triplicate.

### 5.3. Construction of cDNA Libraries and RNA Sequencing

Total RNA extraction, cDNA libraries construction was prepared according to the methods described by Lin et al. [30]. Illumina^®^ HiSeq 4000^TM^ system (San Diego, CA, USA) was used to sequence the cDNA libraries. The original RNA-seq data have been uploaded in the NCBI Sequence Read Archive (SRA) with accession code SUB8228352.

### 5.4. Analysis of Sequence Data

After removing the false reads and deleting the end-sequence with low quality, the reads shorter than 50 bp was discarded. The remaining reads were mapped to the *A. flavus* genome (http://www.ncbi.nlm.nih.gov/nuccore/AAIH00000000). Then, the high-quality reads were assembled into unigenes by using the method described by Grabherr et al. [58]. The transcriptional levels of genes in *A. flavus* were represented using Fragments per kilobase per million mapped fragments (FPKM) values [59]. The mean FPKM of triplicate samples was analyzed by using DEseq software [60] for the differential expression of genes. The significant differential expression genes were identified as log_2_Ratio ≥ 1 and *q* < 0.05 between these compared samples [20]. For the DEGs, gene ontology (GO) functional analysis and Kyoto Encyclopedia of Genes and Genomes (KEGG) pathway enrichment were conducted using FungiFun and KAAs, respectively [61,62,63].

### 5.5. RT-qPCR Analysis of AFs Biosynthesis Genes

All genes in the AFs biosynthesis cluster were chosen for RT-qPCR validation of the RNA-Seq results according to the methods described by Ren et al. [29]. All the data generated from real-time PCR were analyzed using SPSS software version 16.0 with one-way ANOVA method. The significance level of 0.05 has been indicated with lowercases. The gene was defined as significantly up- or down-regulated only if the relative expression level was more than two-fold and showed significant at 0.05 level compared to the control group [64].

## Figures and Tables

**Figure 1 toxins-12-00683-f001:**
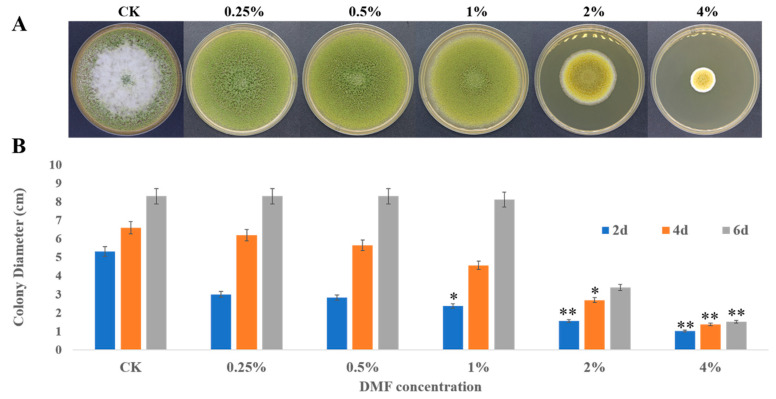
Inhibitory effect of dimethylformamide on fungal growth of *A. flavus* NRRL3357. (**A**) After 6 days of inoculation with *A. flavus* conidia suspension (10^7^), the morphology of *A. flavus* colony on PDA medium under different concentrations (0% to 4%) of dimethylformamide. (**B**) The colony diameter of *A. flavus* treated with dimethylformamide (0 to 4%). CK: control group. Compared with CK, * *p* < 0.05, ** *p* < 0.01.

**Figure 2 toxins-12-00683-f002:**
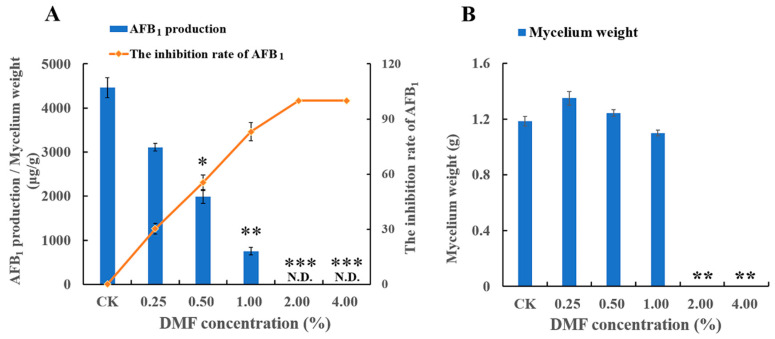
Inhibitory effect of dimethylformamide on AFB_1_ production and fungal growth of *A. flavus* NRRL3357. (**A**) The AFB_1_ production of *A. flavus* and the inhibition rate of AFB_1_ in YES broth at 120 h post-treatment. (**B**) The mycelium weight of *A. flavus* in YES broth at 120 h post-treatment. CK: control group. Compared with CK, * *p* < 0.05, ** *p* < 0.01, *** *p* < 0.001.

**Figure 3 toxins-12-00683-f003:**
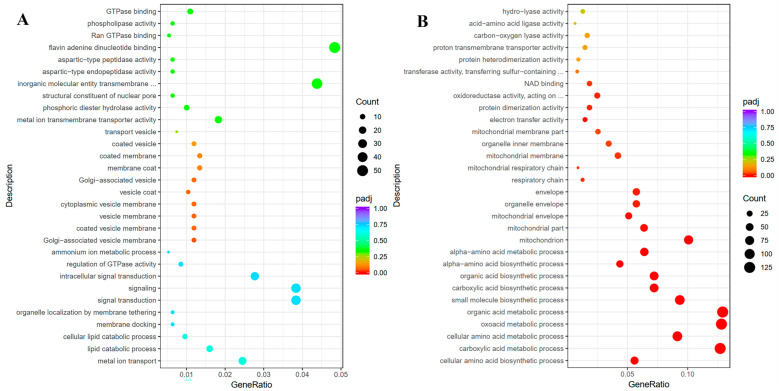
Go functional enrichment of up-regulated (**A**) and down-regulated (**B**) differentially expression genes (DEGs) with 1% dimethylformamide. The ordinate means the -log_10_ of the control and 1% dimethylformamide treatment. The size of the plot represents the number of DEGs in one GO term; the color of the plot close to red represents more significant enrichment.

**Figure 4 toxins-12-00683-f004:**
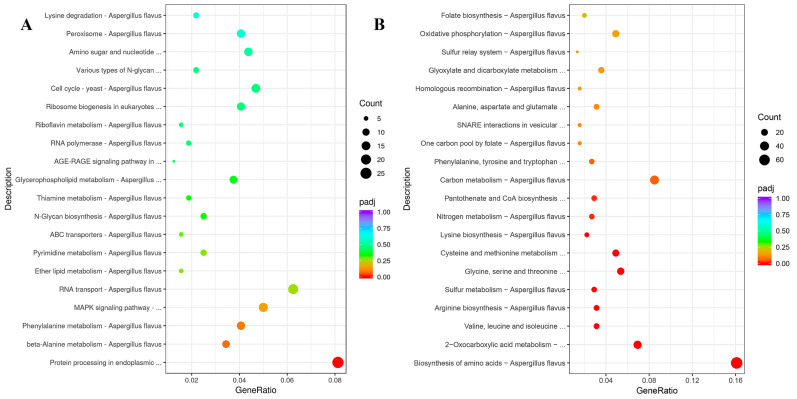
Kyoto Encyclopedia of Genes and Genomes (KEGG) enrichment of up-regulated (**A**) and down-regulated (**B**) DEGs with 1% dimethylformamide. The ordinate represents the KEGG classification. The size of the plot represents the number of DEGs; the color of the plot close to red represents more significant enrichment in one KEGG term.

**Figure 5 toxins-12-00683-f005:**
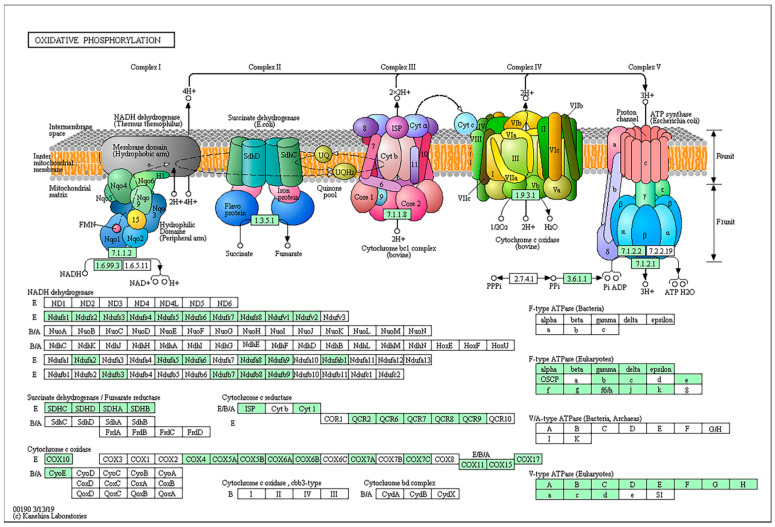
The diagram of oxidative phosphorylation disorder including all enzymes involved in the oxidative phosphorylation, including complexes I, II, III, IV, V. The green box represents the down-regulation expression of the gene.

**Figure 6 toxins-12-00683-f006:**
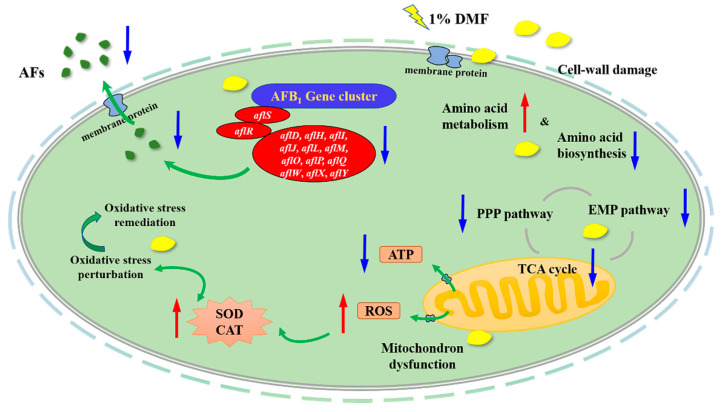
An elementary diagram illustrating the antifungal effect of dimethylformamide act on *A. flavus* NRRL3357. Up- or down-regulation expression of cell substance with dimethylformamide exposure is expressed using red and blue arrow, respectively.

**Table 1 toxins-12-00683-t001:** Transcriptional level of genes involved in *A. flavus* Pigment, development.

Gene ID	CK * (FPKM)	D1 * (FPKM)	Log_2_ D1/CK	Annotated Gene Function
AFLA_016120	4.06	44.50	3.45	O-methyltransferase family protein
AFLA_016130	4.70	45.64	3.28	hypothetical protein
AFLA_016140	2.25	16.60	2.88	*Arp1* conidial pigment biosynthesis scytalone dehydratase
AFLA_006180	0.84	1.08	0.34	*Arb2*/brown2 conidial pigment biosynthesis oxidase
AFLA_009340	2.11	4.24	1.01	*Mod-A* developmental regulator, putative
AFLA_014260	0.62	1.99	1.67	*RodB*/*HypB* conidial hydrophobin
AFLA_098380	0.11	0.13	0.28	*RodA*/*RolA* conidial hydrophobin
AFLA_039530	4.14	19.88	2.26	FluG
AFLA_044790	154.53	520.23	1.75	conidiation-specific family protein
AFLA_044800	18.58	64.49	1.80	conidiation protein Con-6, putative
AFLA_046990	166.39	188.53	0.18	*StuA* APSES transcription factor
AFLA_018340	147.87	136.85	−0.11	*GpaA*/*FadA* G-protein complex alpha subunit
AFLA_081490	36.64	23.97	−0.61	*Gda1/VelB* nucleoside diphosphatase
AFLA_021090	2.06	4.23	1.04	sporulation associated protein
AFLA_024890	25.68	32.99	0.36	*Fsr1*/*Pro1* cell differentiation and development protein
AFLA_029620	2.75	4.35	0.66	*AbaA* transcription factor
AFLA_026900	17.70	32.20	0.86	*VosA* developmental regulator
AFLA_066460	135.21	112.55	−0.26	*VeA* developmental regulator
AFLA_033290	40.16	27.03	−0.57	*LaeA* regulator of secondary metabolism
AFLA_134030	25.77	16.89	−0.61	developmental regulator *FlbA*
AFLA_136410	158.96	139.49	−0.19	transcriptional regulator *Medusa*
AFLA_137320	122.34	73.75	−0.73	C_2_H_2_ conidiation transcription factor *FlbC*
AFLA_052030	11.14	17.11	0.62	*WetA* developmental regulatory protein
AFLA_071090	291.36	406.28	0.48	*EsdC* GTP-binding protein
AFLA_079710	54.76	57.87	0.08	*HymA* conidiophore development protein
AFLA_080170	5.12	7.75	0.60	*FlbD* MYB family conidiophore development protein, putative
AFLA_082850	1.68	2.20	0.39	*BrlA* C_2_H_2_ type conidiation transcription factor
AFLA_083110	34.55	50.12	0.54	conidiation-specific protein (Con-10), putative
AFLA_101920	8.41	14.18	−0.75	*FluG* extracellular developmental signal biosynthesis protein
AFLA_131490	62.27	62.35	0.00	conserved hypothetical protein

* CK = Control; D1 = 1% dimethylformamide.

**Table 2 toxins-12-00683-t002:** Transcriptional level of genes involved in the biosynthesis of Aflatrem (#15), Aflatoxins (#54), and Cyclopiazonic Acid (#55).

Cluster ID	Gene ID	CK * (FPKM)	D1 * (FPKM)	Log_2_ D1/CK	Annotated Gene Function
15	AFLA_045460	2.99	0.90	−1.73	MFS multidrug transporter, putative
15	AFLA_045470	0	0	NA	FAD dependent oxidoreductase, putative
15	AFLA_045480	0.06	0.07	0.20	dimethylallyl tryptophan synthase, putative
15	AFLA_045490	0	0.03	Up	hybrid PKS/NRPS enzyme, putative
15	AFLA_045500	0.04	0.07	0.85	cytochrome P450, putative
15	AFLA_045510	0	0.02	Up	integral membrane protein
15	AFLA_045520	0	0.07	Up	integral membrane protein
15	AFLA_045530	0	0	NA	hypothetical protein
15	AFLA_045540	0.04	0.06	0.76	cytochrome P450, putative
15	AFLA_045550	3.64	6.43	0.82	hypothetical protein
15	AFLA_045560	4.57	6.63	0.54	carboxylic acid transport protein
15	AFLA_045570	2.23	0.62	−1.85	acetyl xylan esterase, putative
54	AFLA_139100	6.51	7.62	0.23	*aflYe*/*orf*/Ser -Thr protein phosphatase family protein
54	AFLA_139110	5.18	7.60	0.55	*aflYd*/*sugR*/sugar regulator
54	AFLA_139120	3.94	7.22	0.87	*aflYc*/*glcA*/glucosidase
54	AFLA_139130	4.60	5.98	0.38	*aflYb*/*hxtA*/putative hexose transporter
54	AFLA_139140	2.68	1.05	−1.34	*aflYa*/*nadA*/NADH oxidase
54	AFLA_139150	9.79	2.77	−1.82	*aflY*/*hypA*/*hypP*/hypothetical protein
54	AFLA_139160	10.32	4.95	−1.06	*aflX*/*ordB*/monooxygenase/oxidase
54	AFLA_139170	15.59	5.08	−1.62	*aflW*/*moxY*/monooxygenase
54	AFLA_139180	11.52	5.79	−0.99	*aflV*/*cypX*/cytochrome P450 monooxygenase
54	AFLA_139190	10.62	5.74	−0.89	*aflK*/vbs/VERB synthase
54	AFLA_139200	3.91	1.43	−1.45	*aflQ*/*ordA*/*ord-1*/oxidoreductase/cytochrome P450 monooxigenase
54	AFLA_139210	16.72	5.02	−1.73	*aflP*/*omtA*/*omt-1*/O-methyltransferase A
54	AFLA_139220	27.50	8.70	−1.66	*aflO*/*omtB*/*dmtA*/O-methyltransferase B
54	AFLA_139230	1.47	0.41	−1.81	*aflI*/*avfA*/cytochrome P450 monooxygenase
54	AFLA_139240	6.38	3.60	−0.82	*aflLa*/*hypB*/hypothetical protein
54	AFLA_139250	10.04	4.37	−1.19	*aflL*/*verB*/desaturase/P450 monooxygenase
54	AFLA_139260	6.54	3.59	−0.86	*aflG*/*avnA*/*ord-1*/cytochrome P450 monooxygenase
54	AFLA_139270	176.54	181.24	0.03	*aflNa*/*hypD*/hypothetical protein
54	AFLA_139280	4.94	4.62	−0.09	*aflN*/*verA*/monooxygenase
54	AFLA_139290	13.89	7.10	−0.96	*aflMa*/*hypE*/hypothetical protein
54	AFLA_139300	55.63	17.29	−1.68	*aflM*/*ver-1*/dehydrogenase/ketoreductase
54	AFLA_139310	15.86	8.02	−0.98	*aflE*/*norA*/aad/adh-2/NOR reductase/dehydrogenase
54	AFLA_139320	33.02	13.07	−1.33	*aflJ*/*estA*/esterase
54	AFLA_139330	28.77	13.33	−1.10	*aflH*/*adhA*/short chain alcohol dehydrogenase
54	AFLA_139340	108.53	90.40	−0.26	*aflS*/pathway regulator
54	AFLA_139360	76.49	57.18	−0.41	*aflR*/*apa-2*/*afl-2*/transcription activator
54	AFLA_139370	8.47	7.73	−0.13	*aflB*/*fas-1*/fatty acid synthase beta subunit
54	AFLA_139380	6.89	7.99	0.21	*aflA*/*fas-2*/*hexA*/fatty acid synthase alpha subunit
54	AFLA_139390	39.85	19.86	−1.00	*aflD*/*nor-1*/reductase
54	AFLA_139400	13.55	8.20	−0.72	*aflCa*/*hypC*/hypothetical protein
54	AFLA_139410	10.41	6.66	−0.64	*aflC*/*pksA*/*pksL1*/polyketide synthase
54	AFLA_139420	130.55	123.18	−0.08	*aflT*/*aflT*/transmembrane protein
54	AFLA_139430	21.56	16.39	−0.39	*aflU*/*cypA*/P450 monooxygenase
54	AFLA_139440	22.49	16.67	−0.43	*aflF*/*norB*/dehydrogenase
55	AFLA_139460	485.20	216.59	−1.16	MFS multidrug transporter, putative
55	AFLA_139470	64.20	249.28	1.95	FAD dependent oxidoreductase, putative
55	AFLA_139480	155.11	556.66	1.84	dimethylallyl tryptophan synthase, putative
55	AFLA_139490	0.95	6.29	2.72	hybrid PKS/NRPS enzyme, putative

* CK = Control; D1 = 1% dimethylformamide; NA = Not applicable, means the FKPM value of the gene in CK group and D1 group were both zero; UP means the FKPM value of the gene in CK group was zero and the transcriptional level of the gene in D1 group was up-regulated compared with CK group.

**Table 3 toxins-12-00683-t003:** Transcriptional activity of genes involved in *A. flavus* cell wall.

Gene ID	CK * (FPKM)	D1 * (FPKM)	Log_2_ D1/CK	Annotated Gene Function
AFLA_038420	0.02	0.59	4.68	endo-chitosanase B
AFLA_024770	0.89	4.22	2.25	symbiotic chitinase, putative
AFLA_023460	5.01	17.45	1.80	alpha-1,3-glucan synthase Ags1
AFLA_134100	0.05	0.09	0.76	alpha-1,3-glucan synthase Ags2
AFLA_052800	293.07	314.70	0.10	1,3-beta-glucan synthase catalytic subunit FksP
AFLA_041060	0.05	0.01	−2.88	cell wall associated protein, putative
AFLA_104680	0.05	0.01	−2.27	class V chitinase ChiB1
AFLA_013280	0.75	0.22	−1.76	class V chitinase, putative
AFLA_031380	88.88	34.17	−1.38	class V chitinase, putative
AFLA_054470	0.31	0.25	−0.30	class V chitinase Chi100
AFLA_114760	24.13	10.06	−1.26	chitin synthase B
AFLA_086070	0.02	0.01	−0.94	chitin synthase, putative
AFLA_067530	43.44	29.37	−0.56	chitin biosynthesis protein (Chs7), putative
AFLA_137200	1.12	0.81	−0.47	chitin synthase, putative
AFLA_013690	78.45	58.64	−0.42	chitin synthase C
AFLA_091300	62.44	50.43	−0.31	chitin biosynthesis protein (Chs5), putative
AFLA_052780	5.03	3.14	−0.68	cell wall glucanase (Scw4), putative
AFLA_096680	1.53	0.41	−1.89	glucan endo-1,3-alpha-glucosidase agn1 precursor, putative
AFLA_095680	1.08	0.34	−1.67	alpha-1,3-glucanase, putative
AFLA_029950	6.84	2.55	−1.43	endo-1,3(4)-beta-glucanase, putative
AFLA_045290	570.20	213.72	−1.42	extracellular endoglucanase/cellulase, putative
AFLA_102640	1.53	0.60	−1.36	exo-beta-1,3-glucanase, putative
AFLA_053390	1894.71	823.30	−1.20	GPI-anchored cell wall beta-1,3-endoglucanase EglC
AFLA_068300	4467.43	2012.10	−1.15	1,3-beta-glucanosyltransferase Bgt1
AFLA_129440	704.92	365.98	−0.95	1,3-beta-glucanosyltransferase, putative
AFLA_034920	4.00	2.11	−0.92	glucan endo-1,3-alpha-glucosidase agn1 precursor, putative
AFLA_058480	4287.31	2388.65	−0.84	1,3-beta-glucanosyltransferase Gel1
AFLA_087870	34.41	20.02	−0.78	Endoglucanase, putative
AFLA_111970	3.86	2.26	−0.77	Endoglucanase, putative
AFLA_126410	4.24	2.61	−0.70	endoglucanase-1 precursor, putative
AFLA_052780	5.03	3.14	−0.68	cell wall glucanase (Scw4), putative

* CK = Control; D1 = 1% dimethylformamide.

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
