# Peer review of "Dimethylformamide Inhibits Fungal Growth and Aflatoxin B1 Biosynthesis in Aspergillus flavus by Down-Regulating Glucose Metabolism and Amino Acid Biosynthesis"

_toxins, 2020, doi:10.3390/toxins12110683_

Round 1

Reviewer 1 Report

In this manuscript, the authors discovered that DMF had an inhibitory effect against A. flavus with a strong capacity to reduce AFs production. In addition, the inhibition mechanism of DMF on AFs production was revealed by the transcriptional expression analysis of genes referred to AFs biosynthesis. The manuscript is well written and developed. Actually, this article presents relevant information that could be of interest for readers in the field of mycology and mycotoxicology with a wide range of applications. It is worthy to mention that significant results are shown in this manuscript but the applicability of this study should be introduced and clarified.

- Line 2 : …and oxidative…

-Line 29: You didn’t mention anything about the carcinogenicity of AFB1, you should specify that AFB1 is classified as type 1 by IARC.

- Line 33: What is the maximum tolerable limits for AFs in crops, etc. you should specify it based on FDA and/or EU commission norms.

-Unfortunately, authors did not test the toxicity of DMF at the used concentrations on cells specially that we are dealing with foods such as crops and other cereal grains. Information about its toxicity should be mentioned in the manuscript. According to CDC, DMF is a potent liver toxin, it may cause different health problems (abdominal pain, nausea, vomiting, skin problems, etc.) and the decision regarding its association with human cancer is not conclusive yet. A clear explanation about its toxicity should be written and an indication of the absence of any toxicity at the tested concentrations have to be mentioned. Even in case there was no clear evidence about its toxicity at these levels a clear indication about conducting later this study should be included.  

- There was no information about how you are planning to apply DMF at an agricultural and industrial level (during harvest by spraying plants or post-harvest, which stage…) and if DMF is only used to inhibit or it may be used to prevent fungal contamination. I think a small section about all these issues can be added.

- There was no indication about the stability of DMF, especially that DMF can be easily photo-chemically degraded. Thus, how you are planning to surpass this problem with plants that needs a specific environment (sun, air…) to grow. Maybe applying DMF directly before storage of crops can be a potential solution for this issue or other solutions may be suggested in order to test and increase DMF stability in suspensions.

Author Response

Response to Reviewer 1 Comments

Point 1: In this manuscript, the authors discovered that DMF had an inhibitory effect against A. flavus with a strong capacity to reduce AFs production. In addition, the inhibition mechanism of DMF on AFs production was revealed by the transcriptional expression analysis of genes referred to AFs biosynthesis. The manuscript is well written and developed. Actually, this article presents relevant information that could be of interest for readers in the field of mycology and mycotoxicology with a wide range of applications. It is worthy to mention that significant results are shown in this manuscript but the applicability of this study should be introduced and clarified.

Response 1: Thank you very much for your recognition to our manuscript. In particular, thank you very much for your critical and valuable comments. The applicability of this study was added at Key Contribution in lines 24-27 of the revised manuscript.

Point 2: - Line 20: …and oxidative…

Response 2: As per your suggestions, the sentence of line 20-21 “amino acid biosynthesis, oxidative phosphorylation.” was changed to “amino acid biosynthesis and oxidative phosphorylation” in the revised manuscript.

Point 3: -Line 29: You didn’t mention anything about the carcinogenicity of AFB1, you should specify that AFB1 is classified as type 1 by IARC.

Response 3: As per your suggestion, we added the relevant content about the carcinogenicity of AFB1. The detailed content “AFs were classified as a Class I carcinogens by the International Agency for Research on Cancer [5,6]” was added in lines 36-37 of the revised manuscript.

Point 4: - Line 33: What is the maximum tolerable limits for AFs in crops, etc. you should specify it based on FDA and/or EU commission norms.

Response 4: As per your suggestion, the maximum tolerable limitation of AFs in crops and feeds has been specified based on FDA and EU commission norms in lines 39-42 of the revised manuscript.

Point 5: -Unfortunately, authors did not test the toxicity of DMF at the used concentrations on cells specially that we are dealing with foods such as crops and other cereal grains. Information about its toxicity should be mentioned in the manuscript. According to CDC, DMF is a potent liver toxin, it may cause different health problems (abdominal pain, nausea, vomiting, skin problems, etc.) and the decision regarding its association with human cancer is not conclusive yet. A clear explanation about its toxicity should be written and an indication of the absence of any toxicity at the tested concentrations have to be mentioned. Even in case there was no clear evidence about its toxicity at these levels a clear indication about conducting later this study should be included.

Response 5: Thank you very much for your critical and valuable comments. As per your suggestions, we added the relevant content to explain its toxicity in line 47-49 of revised manuscript. In our study, some essential and plant hormone are used to inhibit fungal growth and mycotoxins production. However, some fungicides and compounds have low solubility in general solvents. As a universal solvent, dimethylformamide (DMF) is effective to dissolve high-efficiency antifungal agents. If we use DMF as solvents of antifungal agents, to determine the inhibitory effect and mechanism of DMF on A. flavus growth and AFs production is necessary for basic research. Due to the toxicity of DMF, we will not directly use DMF in food industry. In addition, we will assess the toxicity of DMF in the future study.

Point 6: - There was no information about how you are planning to apply DMF at an agricultural and industrial level (during harvest by spraying plants or post-harvest, which stage…) and if DMF is only used to inhibit or it may be used to prevent fungal contamination. I think a small section about all these issues can be added.

Response 6: Thank you very much for your valuable comments. We added the application of dimethylformamide in line 58-60 of revised manuscript. In terms of potential agricultural applications, DMF may synergistically inhibit fungi with new antifungal agents before harvest by spraying plants. The main purpose of this study is to reveal the inhibitory mechanism of action of DMF on AFs biosynthesis at the transcriptomic level and provide a basis for synergistic antifungal and antitoxigenic effect between DMF as a solvent and other fungicides.

Point 7: - There was no indication about the stability of DMF, especially that DMF can be easily photo-chemically degraded. Thus, how you are planning to surpass this problem with plants that needs a specific environment (sun, air…) to grow. Maybe applying DMF directly before storage of crops can be a potential solution for this issue or other solutions may be suggested in order to test and increase DMF stability in suspensions.

Response 7: Thank you very much for your valuable comments. The main purpose of this study is to reveal the inhibitory mechanism of action of DMF on AFs biosynthesis at the transcriptomic level and provide a basis for synergistic antifungal and antitoxigenic effect between DMF as a solvent and other fungicides. So, in the present study, we did not consider the stability of DMF.

Taking into account the stability of DMF, we suggest applying DMF during growth stage of crops. If we want to use DMF in agricultural industry, we will develop method to surpass this problem.

Reviewer 2 Report

The rationale for the work is not well explained: the scientific challenges or the knowledge gaps are not explained or are unrelated to the hypothesis. Consequently, the reader finds it difficult to understand the purpose of the study.

Dimethylformamide (DMF) is a common solvent for chemical reactions but, as far as I know, it is not approved for use in the food industry in the production of foodstuffs and food ingredients. DMF is classified as Group 2A probably carcinogenic to humans by the IARC (2018). The practical interest of studying the effect of DMF on fungal growth and mycotoxin production and the mechanism of action are unclear for the agrifood sector. The direct use of DMF (straight, diluted or as additive) on crops or foods as a detoxifying agent or preventive measure may be questionable for safety reasons. Maybe the reasons for the study are others but they are not explained anyway.

Introduction

Many references are unrelated and in the wrong place. The references to previous works should be very precise throughout the entire paper.

Lines 28-29: Reference 3 is misplaced here as it deals with DMF not with aflatoxins.

Lines 29-31: Reference 4 is not adequate to support the sentence about aflatoxin-susceptible crops. In addition, the sentence is repetitive (peanut, maize, rice repeated twice in the same sentence).

Lines 31-33: Reference 5 is not adequate to support the sentence about aflatoxin toxicity. In addition, the toxic mechanism of action of aflatoxins is not usually described as accumulative.

Lines 36-37: References 7 and 8 are not adequate to support the use of DMF in industrial or medical fields.

The introduction must be supported by precise and relevant references to previous work.

The general background of the study is not well explained, as many references are inappropriate. Is the purpose of the study related to the agricultural field?

Material and methods

2.2. Determination of Fungal Growth and AFB1 Production: the description of control experiments is missing.

Line 70: what is meant by ‘the requisite concentrations’. The terms MIC and MFC are generally used.

Figure 1: what is CK? Control group?

Author Response

Response to Reviewer 2 Comments

Point 1: The rationale for the work is not well explained: the scientific challenges or the knowledge gaps are not explained or are unrelated to the hypothesis. Consequently, the reader finds it difficult to understand the purpose of the study.

Response 1: Thank you very much for your critical and valuable comments. In fact, we discovered the antifungal effect of dimethylformamide (DMF) in the process of studying naphthaleneacetic acid, a new fungicide. Because naphthaleneacetic acid is hardly soluble and has low solubility in conventional solvents, dimethylformamide (DMF) was selected as the solvent. In the above experiment, we found DMF could inhibit fungal growth, AFB1 production and the expression of aflatoxin biosynthetic genes. So, it is necessary to eliminate the influence of solvents, so this experiment was designed. For the application of DMF, it can synergistically inhibit fungi with new antifungal agents. We added some sentences to explain the purpose of the study in Lines 50-59 of the revised manuscript.

Point 2: Introduction

Many references are unrelated and in the wrong place. The references to previous works should be very precise throughout the entire paper.

Lines 28-29: Reference 3 is misplaced here as it deals with DMF not with aflatoxins.

Response 2: Thank you very much for your valuable comments. As per your suggestion, we checked all the references carefully, and the references which not related were deleted.

As per your suggestions, we deleted the reference 3 (Li et al., 2019) in the original manuscript.

Point 3: Lines 29-31: Reference 4 is not adequate to support the sentence about aflatoxin-susceptible crops. In addition, the sentence is repetitive (peanut, maize, rice repeated twice in the same sentence).

Response 3: As per your suggestions, we deleted the repeated phrases and reorganized this sentence, added new references 3 (Luttfullah et al., 2011) and reference 4 (Atayde et al., 2012) in lines 41-44 of revised manuscript to replace original reference 4 (Nasir et al., 2002).

Point 4: Lines 31-33: Reference 5 is not adequate to support the sentence about aflatoxin toxicity. In addition, the toxic mechanism of action of aflatoxins is not usually described as accumulative.

Response 4: As per your suggestion, we deleted the original reference 5 (Khaneghah et al., 2018) and added the new reference 9 and 10 in revised manuscript. The reorganized sentence was added in Lines 41-50 to explain the aflatoxin toxicity.

Point 5: Lines 36-37: References 7 and 8 are not adequate to support the use of DMF in industrial or medical fields.

Response 5: As per your suggestion, we deleted sentence about the use of DMF in industrial or medical fields and the original reference 7 (Nomiyama et al., 2001) and 8 (Nomiyama et al., 2001). In this part, we rephrased some sentences to explain the purpose of this study.

Point 6: The introduction must be supported by precise and relevant references to previous work.

Response 6: As per your suggestion, we sorted out the relevant content of introduction, deleted inappropriate references and added some supporting references in the revised manuscript.

Point 7: The general background of the study is not well explained, as many references are inappropriate. Is the purpose of the study related to the agricultural field?

Response 7: Thank you very much for your valuable comments. As per your suggestion, we added the necessity of this study in line 50-60. Some fungicides have low solubility in general solvents. As a universal solvent, dimethylformamide (DMF) is effective to dissolve high-efficiency antifungal agents. Therefore, to determine the inhibitory effect and mechanism of DMF on A. flavus are necessary for basic research. In terms of agricultural applications, this research may provide a basis for synergistic antifungal effect between DMF and new fungicides.

Point 8: Material and methods

2.2. Determination of Fungal Growth and AFB1 Production: the description of control experiments is missing.

Line 70: what is meant by ‘the requisite concentrations’. The terms MIC and MFC are generally used.

Response 8: As per your suggestion, we added the description of the control group in line 368 of the revised manuscript. The requisite concentrations are the concentrations 0.25%, 0.5%, 1.0%, 2.0% and 4.0% DMF. We deleted the phrase “requisite” in line 380 of revised manuscript. The terms MIC and MFC are not suitable to be used for the above concentrations.

Point 9: Material and methods

2.2. Determination of Fungal Growth and AFB1 Production: the description of control experiments is missing.

Figure 1: what is CK? Control group?

Response 9: Yes, CK is control group. As per your suggestion, we added more details about the control group in lines 381 of the revised manuscript.

Round 2

Reviewer 2 Report

The authors have adequately addressed all questions raised. The present revised manuscript has been improved and can be accepted in present form.